# Nitrogen use efficiency in bread wheat: Genetic variation and prospects for improvement

Suma S. Biradar[1], Mahalaxmi K. Patil[2], S. A. Desai[2]*, Sanjay K. Singh[3]*, V. Rudra Naik[4], Kumar Lamani[5], Arun K. Joshi[6]

**1** AICRP on Wheat, MARS, University of Agricultural Sciences, Dharwad, India, **2** Department of Genetics & Plant Breeding, University of Agricultural Sciences, Dharwad, India, **3** Genetics Division, ICAR-Indian Agricultural Research Institute, New Delhi, India, **4** Directorate of Research, University of Agricultural Sciences, Dharwad, India, **5** Department of Agronomy, University of Agricultural Sciences, Dharwad, India, **6** CIMMYT- India, New Delhi, India

* desaisa@uasd.in (SAD); Sanjay.Singh4@icar.gov.in (SKS)

**Data Availability Statement:** All relevant data are within the paper.

**Funding:** The author(s) received no specific funding for this work.

## Abstract

Nitrogen (N) is one of the primary macronutrients required for crop growth and yield. This nutrient is especially limiting wheat yields in the dry and low fertile agro-ecologies having low N in the root zone soil strata. Moreover, majority of farmers in India and South Asia are small to marginal with meagre capacity to invest in costly nitrogen fertilizers. Therefore, there is an immense need to identify lines that use nitrogen efficiently. A set of 50 diverse wheat genotypes consisting of indigenous germplasm lines (05), cultivars released for commercial cultivation (23) and selected elite lines from CIMMYT nurseries (22) were evaluated in an alpha-lattice design with two replications, a six-rowed plot of 2.5m length for 24 agro morphological, physiological and NUE related traits during two consecutive crop seasons in an N-depleted precision field under two different N levels of 50%-N50 (T1) and 100%-N100 (T2) of recommended N, i.e., 100 kg/ha. Analysis of variance revealed significant genetic variation among genotypes for all the traits studied. About 11.36% yield reduction was observed at reduced N levels. Significant correlations among NUE traits and yield component traits were observed which indicated pivotal role of N remobilization to the grain in enhancing yield levels. Among N-insensitive genotypes identified based on their yielding ability at low N levels, UASBW13356, UASBW13358, UASBW13354, UASBW13357 and KRL1-4 showed their inherent genotypic plasticity toward N application. The genotypes with more yield and high to moderate NUtE can be used as parents for the breeding of N efficient genotypes for marginal agro-ecologies. Low N tolerant genotypes identified from the current investigation may be further utilized in the identification of genomic regions responsible for NUE and its deployment in wheat breeding programs. The comprehensive data of 24 traits under different nitrogen levels for diverse genotypes from India and global sources (mainly CIMMYT) should be useful for supporting breeding for NUE and thus will be of great help for small and marginal farmers in India and South Asia.

**Competing interests:** The authors have declared that no competing interests exist.

## 1. Introduction

Nitrogen (N) is an essential nutrient for optimizing plant growth and reproduction and therefore, applying N fertilizer is an essential practice to secure productivity in diverse cropping systems. Plants take N from both atmospheric air and soil minerals, but crop plants are inefficient in the acquisition and utilization of applied nitrogen. The efficient use of N by plants is determined by their ability to absorb naturally available N or applied N fertilizers. The current global nitrogen (N) demand stands at about 110.19 million metric tons with a projected annual increase of ~1.3% in the future [1] which equates to global N fertilizer consumption of more than 110 Mt per annum of which half of the total is being used for the production of major cereal crops i.e., maize, rice, and wheat [2].

Indian agriculture consumes over 17 million tons of N fertilizer per year. Studies have shown that more than 50% of applied N fertilizers are unused by crops [3, 4] and this unabsorbed N fertilizer in the soil is lost to the environment by leaching, denitrification, runoff, and atmospheric release through volatilization and becomes a major environmental concern [5–9]. Besides, an excess usage of N fertilizer not only decreases the efficiency of nutrient use but also affects the rate of economic returns per unit of chemical fertilizer applied. The effect of negative environmental and economic impacts could be reduced through better agronomic practices and by utilizing N efficient lines with improved nitrogen use efficiency (NUE) [10]. Hence, the development of more N-use efficient cultivars is an important research goal for enhancing future food production capabilities for greater agricultural sustainability [11].

Wheat (*Triticum aestivum* L.) is one of the staple food crops that ensures food and nutritional security for the world's poorest people living in different agro-ecological zones. In the Indian agriculture scenario, wheat is grown under diverse production conditions including low fertile and rainfall dependent central and peninsular areas often facing drought stress during different stages of crop growth. In such areas, farmers apply a high dose of N fertilizer for maximizing grain yield (GY), however excessive usage of N fertilizer leads to a low NUE of the crop along with various environmental hazards. Further, many farmers lack knowledge about the ideal dose of fertilizer to harvest the real yield potential. The majority (90%) of such farmers are small and marginal and do not have capacity to take risk of applying high inputs. Hence, yield improvement under low nitrogen input would be extremely beneficial for economic and environmentally sustainable cultivation of wheat crop.

Similar to grain yield, nitrogen use efficiency (NUE) is also a complex trait, which is associated with various morphological, physiological, molecular, and biochemical changes in plants throughout the life cycle. For a clear understanding of this complex nature, studies of various physiological traits and their close correlation with one or more economically important traits like GY are foremost critical. Nevertheless, it will help in selecting low N tolerant/high yielding lines at different N conditions [12]. In general, plant function is always associated with chlorophyll content, which directly indicates the N status of the leaf [13]. Leaf chlorophyll content and photosynthetic capacity are appropriate benchmarks for identifying high NUE (HNUE) genotypes under low N conditions during field trials [14, 15]. Leaf N status was usually measured by using a hand-held optical chlorophyll meter to monitor the leaf nitrogen status and chlorophyll content.

NUE has several definitions based on agronomic, genetic, or physiological studies [16, 17]. NUE is the efficiency of nitrogen recovery from applied fertilizer, or the N available to the crop, giving rise to 33% efficiency of the crop recovery [3, 17, 18] (Alternatively, it is often considered a productivity index and defined as the yield produced per unit of available N to the crop expressed in terms of kg yield per kg of available N [19, 20]. NUE included two components namely, N uptake efficiency (NUpE) and N utilization efficiency (NUtE) and is

mathematically the product of NUpE x NUtE [21–23]. NUpE is the ability of the plant to take up N from the soil. It is estimated as the ratio of N taken up by the plant per unit of N available from the soil and applied fertilizer and is expressed as kg N (in the crop) per kg N (available). NUtE is the ability to use N to produce gain yield which is estimated as the grain yield produced per unit of N taken up by the plant and is also expressed as kg (grain) per kg N [11, 24].

Studies on genotypic variations under low and recommended N for NUE traits in various crops spurred breeding activities to develop N use-efficient genotypes possessing high yield sustainability under low N conditions [11, 14, 25–32]. There have been continuous improvements in N use efficiency (NUE) of crops over the years along with increase in crop yield [33–36] and extensive utilization of the genetic pool through introgression of landraces and ancestral germplasm is expected to support breeding programs in driving future NUE-based outcomes [36, 37]. The molecular studies identified 333 genomic regions associated to 28 NUE related traits in winter wheat and NUE and its related traits were found to be polygenic in nature and highly influenced by the genetic backgrounds [38].

The high NUE/ N-insensitive genotypes (NIS-top grain yielders) give more or equal GY with minimal application of N fertilizer compared to recommended N fertilizer conditions [39]. The understanding of GY and its associated NUE traits is still lagging in wheat but few studies have explored genotypic variations for NUE at different N levels [40, 41]. Several studies in cereals have suggested an enhanced NUE through improved NUtE under low N conditions [19, 21]. The nitrogen harvest index (NHI) is another useful measure of the efficiency of N use, which is the fraction of total N taken up by the crop that is partitioned to the grain. NHI is independent of yield and uptake efficiency, and a low yielding crop may have a high NHI [42].

The green revolution period in India had focus primarily on the responsiveness of wheat genotypes to high-input supply. In the present scenario of climate change, realizing consistent output (yield and quality) of the genotype under varying levels of nitrogen input is a major challenge to wheat improvement programme. Identifying promising genotypes for enhanced NUE in wheat and estimating its genetic variability can provide useful guidance to breeders [41]. Keeping the high yield when N supply is reduced and/or increasing yield when N supply is continuous may be two strategies for NUE enhancement. Nitrogen production costs, environmental pollution owing to nitrate leaching [43, 44], and greenhouse gas volatilization necessitate improvement of wheat nitrogen use efficiency at a reduced N supply. Simultaneously, GY is a complex trait controlled by a network of multiple traits and their associations. Hence, uncovering the genetic basis of GY, and other related NUE traits under low and high N conditions is a prerequisite to understanding the mechanism and for ideal selection of NUE associated traits. Nitrogen use efficient crop varieties can be a great choice for ensuring sustainability in farming systems and meeting future customer requirements, particularly in the face of changing environment caused by climate change. This study provides useful information for uncovering the physiological and genetic basis of GY and its related traits under low N which further facilitates the development of low N stress resilient wheat cultivars.

## 2. Material and methods

### 2.1 Selection of genotypes

The genotypes included in the study consisted of fifty bread wheat genotypes representing high, medium, and low NUE based on multi-location and multi-year evaluation. These genotypes represent indigenous elite lines (DTW2011-56, GW2013-540, MP1293, RAJ4248, UAS323), cultivars released for commercial cultivation (DBW14, DL153-2, GW322, HD2189, HD2733, HD2967, HI1500, HI1530, HPW251, K9107, Kalyansona, KRL1-4, KRL237, MP4010, NP846, PBW175, PBW343, RAJ1972, UAS304, WH147, WH542, WH1021,

WH1022) and selected elite lines from CIMMYT trials/nurseries (UASBW10227, UASBW10453, UASBW11948, UASBW12380, UASBW12758, UASBW12876, UASBW12877, UASBW12878, UASBW13354, UASBW13355, UASBW13356, UASBW13357, UASBW13358, UASBW13359, UASBW13360, UASBW13361, UASBW13362, UASBW13363, UASBW13364, UASBW13365, UASBW13366, UASBW13367). The investigation was carried out for two years during *rabi* 2016–17 and 2017–18 at All India Coordinated Wheat Improvement Project, Main Agricultural Research Station, University of Agricultural Sciences, Dharwad located at 15.48˚N latitude and 74.98˚E longitude and falls in agro-climatic Zone 8 of Karnataka state in Peninsular India.

## 2.2 Experimental design, treatment, and crop management

The experiment was laid out in an Alpha-lattice design at two treatment levels of nitrogen (T1 and T2) with two replications. The $T_1$ consisted of soil N + 50% of the recommended nitrogen dose (RND) of 100 kg N $ha^{-1}$ which is equivalent to 50kg/$ha^{-1}$ whereas $T_2$ consisted of soil N +RND of 100 kg N $ha^{-1}$. The genotypes were randomly allotted to plots in each treatment in each replication and grown in 6 row plots of 2.5 meter length with 20 cm row spacing. The nitrogen fertilizer dose was calculated as per the treatments and was applied to each plot. The soil samples from the experimental sites were collected from 0 to 30 cm soil depth before the start of the field experiment and after harvest of the crop and N was analyzed using standard procedure. All other recommended agronomic package of practices was adopted to raise a good crop.

## 2.3 Observations

Data on 24 agro-morphological, physiological, and NUE related traits were recorded from phase 41 to phase 92 of the Zadok's scale [45] for wheat plants in the four central rows of each plot. The agro-morphological traits included, days to heading, days to flowering, plant height (cm), number of productive tiller per meter row length, spike length (cm), awn length (cm), spikelet number per spike, grains number per spike, 1000-grains weight (g), grain yield (q ha-1)biomass yield (q ha-1) and harvest index (%); Physiological traits namely, chlorophyll content (SPAD) and normalized difference vegetation index (NDVI) were recorded at booting (SPAD 1, NDVI-1), anthesis (SPAD-2, NDVI-2) and grain filling (SPAD-3, NDVI-3) stages. The chlorophyll content was measured by a Chlorophyll meter or Spad-meter and NDVI was measured by the Trimble GreenSeeker handheld crop sensor using standard procedures. Nitrogen use efficiency (NUE) and its related parameters like nitrogen uptake efficiency (NUpE), nitrogen utilization efficiency (NUtE), nitrogen harvest index (NHI), and total nitrogen uptake (TNUp) were worked out according to [19, 46]. NHI was calculated as [Grain N/ (Grain N + straw N)] *100 and expressed in percentage. The grain N (%) and straw N (%) contents were analyzed by the Micro-Kjeldahl digestion method as per the FAO guidelines for plant nutrient analysis [47]. TNUp (kg N ha-1) was determined as the sum of nitrogen in straw and grain at harvest. NUpE was calculated as total N uptake/crop N supply where crop N supply includes fertilizer N + soil mineral N at planting. NUtE was calculated as grain yield/ total N uptake. NUE was measured as Grain yield/ total nitrogen supply from the soil and applied fertilizer. The values for NUE, NUtE and NUpE were expressed in kg $kg^{-1}$.

## 2.4 Statistical analysis

Before pooling the data across environments, the ANOVA assumption was tested for its homogeneity using the Bartlet test [48]. The ratio of the highest error means square and the smallest error mean square value was compared with the F table and it showed non-significant

results for all the traits hence, two years of data across environments were pooled. From pooled data, the analysis of variance (ANOVA) was done [49]. Critical difference was estimated to see significant differences among the genotypes for various studied traits. The genotypic and phenotypic coefficients of variation [50] and heritability [51, 52] were worked out and the traits were categorized as low, medium and high based on genetic parameters [53, 54]. The correlation among all the traits studied under both the nitrogen levels was worked out using Karl Pearson's simple correlation coefficient method.

For understanding the effect of nitrogen doses on different genotypes concerning yield as well as NUE, various indexes namely, stress susceptibility index [55], percentage reduction [56] and yield stability index [57] were also estimated to identify promising genotypes. As grain yield and NUE measurement are reflective of each other in totality, grain yield was used to estimate these indices.

# 3. Results

## 3.1 Trait specific variability among genotypes

The computed mean sum of squares for various traits namely, chlorophyll content at booting stage (CC-1), anthesis stage (CC-2), grain filling stage (CC-3), NDVI at booting stage (NDVI-1), anthesis stage (NDVI-2), grain filling stage (NDVI-3), days to heading (DH), days to maturity (DM), plant height (PH), number of productive tiller per meter row length (TPM), spike length (SL), awn length (AL), number of spikelet's per spike (SPS), number of grains per spike (GPS), thousand grains weight (TGW), biomass yield (BMY), harvest index (HI), grain protein content (GPC), nitrogen harvest index (NHI), total nitrogen uptake (TNUp), nitrogen uptake efficiency (NUpE), nitrogen utilization efficiency (NUtE) and nitrogen use efficiency (NUE) showed that genotypes differed significantly for all the traits which indicated the existence of variation among genotypes for various traits studied. A wide range was observed for all the traits evaluated under T1 (soil N + 50 kg Nha$^{-1}$) and T2 (Soil N + 100 kg Nha$^{-1}$) as shown in Table 1. In general, the mean values for all the traits except HI, NHI, NUtE and NUE were higher at full/recommended fertilizer dose as compared to reduced nitrogen dose. A similar trend was also observed for trait specific maximum values where all the traits except NDVI-1, SL, HI and NUE showed higher values under T2 as compared to T1 (S1 Table). DMRT encompasses the calculation of numerical thresholds, facilitating the categorization of the disparity between any two treatment means as either statistically significant or not. Treatment means sharing a common letter are deemed non-significantly different at the 5% level of significance. Among the traits examined, NDVI1, AL, HI, NHI, NUpE, and NUE, did not exhibit significant differences at the 5% level of significance (Table 1).

**3.1.1. Yield and agro-morphological traits.** The present investigation included 12 agro morphological traits including grain yield. A wide range was observed for these yield and agro-morphological traits under both the nitrogen levels. In T1 (N50), the average GY of the tested genotypes was 28.65 q/ha and ranged from 16.87 to 38.91q/ha whereas in T2 (N100), GY ranged from 20.40 q/ha to 40.36q/ha with a mean of 32.32 q/ha. Nitrogen in limited condition (N50) resulted in 11.36% reduction in GY compared to N100. A higher mean BMY of 105.9 q/ha with a range of 79.1 to 132.4q/ha was recorded in T2 as compared to 90.8 q/ha mean BMY with range of 73.0 to 108.7q/ha under the T1 condition that showed 14.25% reduction. Interestingly, 1.3% lower HI was observed in T2 (31.5%) as compared to T1 (31.9%) condition. TPM, GPS and TGW are major yield component traits that significantly contribute to higher yield realization. The results showed a higher mean of 97.3 TPM, 50.1 GPS and 37.0g TGW in the T2 condition as compared to 86.0 TPM, 47.0 GPS and 34.1g TGW in the T1

**Table 1. Pooled ANOVA and genetic parameters for morpho-physiological, yield, yield attributes, NUE and NUE related traits in 50 genotypes of bread wheat under T1 (soil N + 50 kg Nha-1) and T2 (Soil N + 100 kg Nha-1).**

| Traits | F statistics (Genotypes at 49 df) | | Mean | | Range | | PCV | | GCV | | $h^2bs$ (%) | | Difference | CD @ 5% T1 | CD @ 5% T2 |
|---|---|---|---|---|---|---|---|---|---|---|---|---|---|---|---|
| | T1 | T2 | T1 | T2 | T1 | T2 | T1 | T2 | T1 | T2 | T1 | T2 | (% of HN) | | |
| CC 1 | 2.54** | 5.22** | 47.70b | 51.63a | 42.95–55.93 | 45.70–59.65 | 6.7 | 6.45 | 4.42 | 5.32 | 43.49 | 67.82 | 7.61*** | 4.89 | 3.84 |
| CC 2 | 3.70** | 6.82** | 49.24b | 57.68a | 42.00–56.45 | 51.70–67.15 | 5.6 | 6.25 | 4.25 | 5.39 | 57.47 | 74.43 | 14.63*** | 3.92 | 3.66 |
| CC 3 | 5.46** | 5.00** | 47.87b | 50.96a | 40.50–56.00 | 45.70–57.83 | 7.03 | 6.12 | 5.84 | 5.00 | 69.05 | 66.71 | 6.06*** | 3.82 | 3.62 |
| NDVI 1 | 4.14** | 3.41** | 0.54a | 0.53a | 0.47–0.61 | 0.48–0.60 | 5.1 | 5.64 | 3.99 | 4.17 | 61.06 | 54.69 | -1.89ns | 0.03 | 0.04 |
| NDVI 2 | 3.48** | 4.67** | 0.57b | 0.60a | 0.50–0.66 | 0.53–0.68 | 6.33 | 5.70 | 4.71 | 4.58 | 55.37 | 64.73 | 5.00*** | 0.05 | 0.04 |
| NDVI 3 | 3.83** | 2.86** | 0.55b | 0.56a | 0.48–0.62 | 0.51–0.65 | 5.43 | 5.49 | 4.16 | 3.81 | 58.61 | 48.15 | 1.79* | 0.04 | 0.04 |
| DH | 14.20** | 22.97** | 59.42b | 62.25a | 50.75–65.25 | 55.25–68.00 | 5.52 | 4.66 | 5.14 | 4.46 | 86.84 | 91.66 | 4.55*** | 2.39 | 1.68 |
| DM | 13.93** | 31.48** | 94.73b | 99.99a | 81.75–104.50 | 91.75–110.50 | 5.25 | 3.84 | 4.88 | 3.72 | 86.6 | 93.84 | 5.26*** | 3.76 | 1.92 |
| PH | 2.21** | 2.28** | 74.41b | 80.02a | 61.48–90.70 | 63.98–93.98 | 9.14 | 9.88 | 5.62 | 6.17 | 37.77 | 39.01 | 7.01*** | 10.89 | 12.51 |
| TPM | 4.50** | 2.76** | 85.95b | 97.32a | 60.00–111.25 | 79.50–121.50 | 15.68 | 11.53 | 12.51 | 7.89 | 63.63 | 46.76 | 11.68*** | 16.39 | 16.46 |
| SL | 2.03** | 3.03** | 7.95b | 8.30a | 6.84–10.07 | 6.92–9.96 | 10.56 | 9.30 | 6.16 | 6.61 | 34.05 | 50.4 | 4.22* | 1.44 | 1.10 |
| AL | 2.28** | 9.41** | 6.27a | 6.72a | 5.43–8.29 | 4.72–8.50 | 12.48 | 14.15 | 7.8 | 12.72 | 39.02 | 80.78 | 6.70ns | 1.23 | 0.84 |
| SPS | 4.82** | 4.37** | 16.44b | 17.8a | 13.20–19.80 | 15.25–20.30 | 9.41 | 5.91 | 7.63 | 4.68 | 65.65 | 62.77 | 7.64*** | 1.83 | 1.29 |
| GPS | 3.68** | 4.56** | 47.00b | 50.1a | 37.60–57.50 | 42.75–57.75 | 9.34 | 6.09 | 7.07 | 4.87 | 57.3 | 64.02 | 6.19*** | 5.98 | 3.67 |
| TGW | 3.71** | 11.53** | 34.10b | 36.99a | 29.13–43.05 | 31.58–46.40 | 10.18 | 8.23 | 7.72 | 7.55 | 57.53 | 84.03 | 7.81*** | 4.59 | 2.46 |
| GY | 5.62** | 7.09** | 28.65b | 32.32a | 16.87–38.91 | 20.40–40.36 | 16.22 | 11.4 | 13.55 | 9.89 | 69.79 | 75.27 | 11.36*** | 5.18 | 3.94 |
| BMY | 4.29** | 4.02** | 90.81b | 105.90a | 72.95–108.71 | 79.13–132.38 | 11.22 | 15.6 | 8.85 | 12.10 | 62.22 | 60.13 | 14.25*** | 12.59 | 21.51 |
| HI | 11.14** | 8.08** | 31.89a | 31.49a | 20.77–52.12 | 16.88–50.01 | 20.47 | 21.61 | 18.71 | 19.08 | 83.53 | 77.98 | -1.27ns | 5.38 | 6.70 |
| GPC | 2.66** | 2.74** | 12.06b | 12.56a | 10.56–13.29 | 10.85–13.70 | 7.49 | 6.81 | 5.04 | 4.65 | 45.35 | 46.5 | 3.98** | 1.34 | 1.26 |
| NHI | 6.84** | 2.22** | 75.83a | 73.79a | 70.03–78.60 | 66.09–78.64 | 2.97 | 4.67 | 2.57 | 2.87 | 74.52 | 37.81 | -2.76*** | 2.29 | 5.80 |
| TNUp | 6.02** | 2.57** | 94.59b | 116.24a | 71.86–113.11 | 88.97–134.78 | 10.81 | 9.10 | 9.15 | 6.04 | 71.54 | 43.97 | 18.63*** | 10.96 | 18.50 |
| NUpE | 6.23** | 2.57** | 0.52a | 0.54a | 0.40–0.63 | 0.42–0.63 | 14.54 | 9.67 | 12.36 | 6.42 | 72.34 | 43.98 | 3.70* | 0.06 | 0.09 |
| NUtE | 4.20** | 5.35** | 30.16a | 28.08b | 22.11–35.15 | 17.95–35.17 | 10.63 | 12.24 | 8.34 | 10.13 | 61.54 | 68.5 | -7.41** | 4.03 | 4.40 |
| NUE | 5.83** | 8.97** | 15.85a | 15.16a | 9.36–21.54 | 9.57–18.95 | 16.54 | 12.02 | 13.91 | 10.75 | 70.73 | 79.95 | -4.55ns | 2.85 | 1.70 |

PCV, GCV- Phenotypic and genotypic coefficients of variation, h2bs- heritability broad sense,CC1, CC2, CC3 & NDVI1, NDVI2, NDVI3- Chlorophyll contents and NDVI at booting, anthesis and grain filling stages, respectively, DH- Days to heading, DM- Days to maturity, PH- Plant height, TPM- Number of productive tiller per meter, SL- Spike length, AL-Awn length, SPS-Number of spikelet's per spike, GPS-Number of grains per spike, TGW-Thousand grain weight, GY: Grain yield, BMY-Biomass yield, HI- Harvest index, GPC-Grain protein content, NHI- Nitrogen harvest index, TNUp-Total nitrogen uptake, NUpE-Nitrogen uptake efficiency, NUtE-Nitrogen utilization efficiency, NUE-Nitrogen use efficiency

Note: ns: not significant.

*, **, and ***: significant at the level of probability $p < 0.05$, $p < 0.01$, and $p < 0.001$, respectively

Note: Means with the same letter are not significantly different at 5% level

condition indicating 11.7, 6.19, and 7.81 percent reductions for TPM, GPS and TGW respectively under T1 condition.

The lower mean values for other morphological traits DH, DM, PH, SL, AL and SPS were observed in T1 with 4.6, 5.3, 7.0, 4.2, 6.7 and 7.6 percent reduction, respectively as compared to mean values for these traits in the T2 condition.

**3.1.2 Physiological parameters.** Chlorophyll content and NDVI values are two physiological parameters that directly indicate the phenotypic nitrogen levels in crop plants based on the greenness of leaves. Chlorophyll content (CC-1, CC-2, CC-3) and NDVI values (NDVI-1, NDVI-2, NDVI-3) were measured at booting, anthesis, and grain filling stages, respectively which also showed a wide range in both conditions. Higher mean values of 51.6, 57.7 and 51.0

chlorophyll content at all three stages were recorded in T2 as compared to 47.7, 49.2 and 47.9 in the T1 condition with a reduction of 7.6%, 14.6% and 6.1%, respectively. Higher chlorophyll content was observed at the anthesis stage in both T1 and T2 conditions. A similar trend was observed for NDVI at anthesis and grain filling stages with 5.0% and 1.8% reduction in the T1 but NDVI at booting stage was a little higher in T1 as compared to T2.

### 3.1.3 NUE associated traits.

Nitrogen based six traits were estimated in the present investigation (Table 1). The NUE in T1 ranged from 9.4 to 21.5 kg grain/kg N (Avg: 15.9 kg grain/kg N) whereas it was with a range of 9.6 to 19.0 kg grain/kg N in T2 (Avg: 15.2 kg grain/kg N) which indicated 4.6% reduction as compared to T1. NUE is mainly dependent on TNUp, NupE and NutE. The results indicated an increase of TNUp with increased application of N fertilizer. It ranged from 71.9 to 113.1 kg N/ha (Avg: 94.6 kg grain/kg N) in T1 and decreased by 18.6% as compared with T2 (N100) where it ranged from 89.0 to 134.8 kg N/ha (Avg: 116.2 kg N/ha). Similarly, NupE was with a range of 0.40 to 0.63 (0.52) and 0.42 to 0.63 (0.54) in T1 and T2, respectively indicating 3.7% reduction in T1. A large variation was observed for NUtE of genotypes which ranged from 22.1 to 35.2 kg grain/kg N (Avg: 30.2 kg grain/kg N) under T1 and 18.0 to 35.2 (Avg: 28.1) in T2. The overall mean showed 7.4% reduction in NUtE in T2 due to poor performance of some of the genotypes which was reflected in lower values of range in T2. The NHI varied from 70.0 to 78.6% (Avg: 75.8%) in the T1 condition, whereas it ranged from 66.1 to 78.6% (Avg: 73.8%) in T2 indicating 2.76% reduction as compared to T1. Grain protein content (GPC) is an important trait depending on N uptake and its utilization and in this study, it was approx. 4.0% higher in T2 as compared to the mean of T1. The GPC ranged from 10.9–13.7% (Avg: 12.6%) in T2 as compared to T1 with a range of 10.6 to 13.3% (Avg: 12.1%).

## 3.2 Genetic parameters

Genetic parameters namely phenotypic (PCV) and genotypic (GCV) coefficients of variation, heritability in the broad sense ($h^2bs$) were worked out as shown in Table 1.

### 3.2.1 Coefficients of variation.

In this study, phenotypic (PCV) and genotypic (GCV) coefficients of variation were estimated in T1 and T2 conditions. In general, higher PCV values were observed as compared to GCV values at both the nitrogen levels. The PCV values ranged from 2.97 (NHI) to 20.47 (HI) under T1 whereas it ranged from 3.84 (DM) to 21.61 (HI) under T2 condition. Harvest index showed a high PCV value, both for T1 (20.47) and T2 (21.61), respectively. The traits TPM (15.68, 11.53), AL (12.48, 14.15), GY (16.22, 11.4), BMY (11.22, 15.6), NUtE (10.63, 12.24) and NUE (16.54, 12.02) have moderate PCV values (10–20) whereas other traits showed low PCV values (<10) under T1 and T2, respectively. Similarly, GCV ranged from 2.57 and 2.81 for NHI to 18.71 and 21.08 for HI under T1 and T2, respectively. Moderate GCV values for HI (18.71, 19.08) and NUE (13.91, 10.75) were observed under both T1 and T2. Moderate GCV values were also observed for TPM (12.51), GY (13.55), and NUpE (12.36) under T1 and for AL (12.72), BMY (12.1), and NUtE (10.13) under T2. Other traits showed low GCV values in T1 and T2.

### 3.2.2 Heritability.

The heritability values in broad sense ranged from 34.05% for SL to 86.84% for DH in T1 and 37.81% for NHI to 93.84% for DM in T2 (Table 1). At reduced nitrogen level (T1), the highest heritability was observed for DH (86.84) followed by DM (86.60) and HI (83.53) whereas it was highest for DM (93.84) followed by DH (91.66), TGW (84.03), and AL (80.78) at full nitrogen dose (T2). Additionally, a high heritability (>60%) was observed for CC-3 (69.1, 66.7), SPS (65.7, 62.8), GY (69.8, 75.3), BMY (62.2, 60.1), NUE (70.7, 80.0) and NUtE (61.5, 68.5) at both T1 and T2 conditions, respectively. Traits NHI (74.5), TNUp (71.5), NUpE (72.3), TPM (63.6) and NDVI-1 (61.1) showed a high heritability under

T1 only whereas for T2, CC-2 (74.4), CC-1 (67.8), NDVI-2 (64.7) and GPS (64.0) showed a high heritability. The plant height showed a low heritability under both T1 (37.8), and T2 (39.0) conditions.

### 3.3 Phenotypic correlation

Character association was studied among genotypes using Karl Pearson's simple correlation coefficient method to identify the interrelation among traits for both the T1 and T2. Nitrogen use efficiency was measured as grain yield divided by available N soil (soil N + fertilizer N) due to which the correlation of nitrogen use efficiency and grain yield at both levels of nitrogen treatments showed similar results. The association between grain yield and NUE under the two nitrogen levels is visually depicted in S1 Fig. Therefore, the correlation values with NUE were estimated as shown in Table 2. The results indicated a positive and highly significant correlated response of nitrogen use efficiency with DM, HI, TGW, GPS, SPS, CC-1, CC-2, CC-3, NDVI-1, NDVI-2, NDVI -3 TNUp, NUpE and NUtE under both T1 and T2 conditions. Similar correlations of NUE with TPM, SL, DH, GPC, PH, TPM, SL and BMY were also observed either in T1 or T2 condition. All the six physiological traits namely CC-1, CC-2, CC-3, NDVI-1, NDVI-2 and NDVI -3 showed a significantly positive correlation with each other. Among NUE related traits, significant and positive correlation of GPC with NHI, TNUp, and NUpE and of TNUp with NUpE and NUtE was observed under both T1 and T2 conditions. For agro-morphological traits, NUE/grain yield showed highly positive correlations with DM, SPS, GPS, TGW and HI under both nitrogen conditions. Traits like SPS, GPS and TGW have a highly significant and positive correlation with each other under both conditions. DM and DH have highly significant and positive associations in T1 as well as T2 conditions. Besides these, highly significant and positive associations of SL with DM, TPM, SPS, GPS, TGW, HI; TPM with SPS, GPS, HI; HI with SPS, GPS, TGW and DH with PH were observed the in T1 condition whereas the similar correlation of DH and DM with SPS and HI were observed in the T2 condition. A highly significant but negative association of HI with BMY was also observed under both nitrogen levels.

The NUE related traits showed a highly significant and positive correlations with yield and its component traits among which GPC with TGW & HI; BMY with TNUp & NUpE and NUtE with SPS, GPS, TGW & HI are prominent ones. All the physiological traits namely chlorophyll content and NDVI at booting, anthesis and grain filling stages showed a highly significant and positive correlations with NUE related traits GPC, TNUp, and NUpE under both conditions. NUtE has a similar association with chlorophyll content at all three stages under the T1 condition and with chlorophyll content and NDVI at the anthesis stage in T2 condition. Considering yield and physiological traits, SPS, GPS, TGW, and HI showed a highly significant and positive associations with all six physiological parameters under both T1 and T2 conditions. Under both conditions, BMY showed significant negative associations with chlorophyll content at all three stages. Significantly positive associations of TPM with all the physiological parameters and SL with chlorophyll content (CC-1, CC-2, CC-3) were observed in T1 only. In T2, significant and positive associations of DH and DM with NDVI and AL with chlorophyll content at different stages were observed. BLUP analysis revealed a positive and weak correlation of GY with GPC (S2A Fig) and NUpE (S2B Fig) under both N levels ($R^2$ = 0.09 at T1 and 0.24 at T2 for GPC and $R^2$ = 0.66 in T1 and T2 ($R^2$ = 0.13 in T2 for NUpE). Furthermore, a moderately positive correlation was observed between GY and NUtE at T1 ($R^2$ = 0.64), while a weaker but positive relationship was observed at T2 conditions ($R^2$ = 0.57) (S2C Fig). Remarkably, the NUE component trait exhibited a perfect positive relationship with grain yield at both T1 and T2 ($R^2$ = 1.00) (S2D Fig).

**Table 2. Phenotypic correlations among different morpho-physiological, yield, yield attributes, NUE and NUE related traits in 50 genotypes of bread wheat under T1 (soil N + 50 kg Nha⁻¹) and T2 (Soil N + 100 kg Nha⁻¹) conditions.**

| T1\T2 | CC1 | CC2 | CC3 | NDVI1 | NDVI2 | NDVI3 | DH | DM | PH | TPM | SL | AL | SPS | GPS | TGW | BY | HI | GPC | NHI | TNUp | NUpE | NUtE | NUE |
|---|---|---|---|---|---|---|---|---|---|---|---|---|---|---|---|---|---|---|---|---|---|---|---|
| CC 1 |  | 0.97** | 0.90** | 0.82** | 0.38** | 0.40** | 0.14 | 0.07 | -0.02 | 0.19 | 0.16 | 0.22* | 0.21* | 0.23* | 0.54** | -0.23* | 0.44** | 0.38** | -0.06 | 0.25* | 0.24* | 0.21* | 0.40** |
| CC 2 | 0.90** |  | 0.97** | 0.38** | 0.39** | 0.42** | 0.16 | 0.10 | 0.02 | 0.15 | 0.17 | 0.21* | 0.25* | 0.27** | 0.57** | -0.24* | 0.47** | 0.42** | -0.06 | 0.25* | 0.24* | 0.26** | 0.45** |
| CC 3 | 0.82** | 0.87** |  | 0.45** | 0.48** | 0.52** | 0.15 | 0.07 | 0.02 | 0.21* | 0.15 | 0.24* | 0.22* | 0.24* | 0.52** | -0.24* | 0.44** | 0.43* | -0.03 | 0.25* | 0.24* | 0.21* | 0.41** |
| NDVI1 | 0.38** | 0.38** | 0.45** |  | 0.83** | 0.88** | 0.35** | 0.34** | -0.08 | -0.01 | 0.08 | 0.19 | 0.31** | 0.31** | 0.36** | 0.08 | 0.24* | 0.36** | -0.01 | 0.45** | 0.44** | 0.11 | 0.43** |
| NDVI2 | 0.40** | 0.39** | 0.48** | 0.56** |  | 0.91** | 0.37** | 0.36** | -0.01 | 0.05 | 0.16 | 0.14 | 0.45** | 0.44** | 0.48** | -0.07 | 0.42** | 0.34** | 0.06 | 0.32** | 0.31** | 0.27** | 0.51** |
| NDVI3 | 0.43** | 0.42** | 0.52** | 0.73** | 0.64** |  | 0.34** | 0.34** | -0.06 | 0.06 | 0.15 | 0.20* | 0.45** | 0.43** | 0.46** | -0.06 | 0.39** | 0.47** | 0.09 | 0.37** | 0.37** | 0.25* | 0.51** |
| DH | 0.12 | 0.11 | 0.06 | 0.35** | 0.37** | 0.34** |  | 0.79** | 0.16 | -0.04 | 0.21* | -0.05 | 0.38** | 0.38** | 0.08 | -0.27** | 0.40** | 0.13 | 0.07 | -0.08 | -0.08 | 0.42** | 0.36** |
| DM | 0.06 | 0.08 | 0.03 | 0.34** | 0.36** | 0.34** | 0.80** |  | -0.02 | -0.12 | 0.18 | 0.00 | 0.26** | 0.24* | 0.05 | -0.09 | 0.29** | 0.14 | 0.28** | 0.01 | 0.01 | 0.41** | 0.40** |
| PH | 0.11 | 0.06 | 0.04 | -0.08 | -0.01 | -0.06 | 0.27** | 0.20* |  | -0.08 | 0.04 | 0.09 | -0.01 | -0.01 | -0.21* | -0.04 | -0.08 | -0.08 | -0.15 | -0.10 | -0.09 | -0.09 | -0.17 |
| TPM | 0.28** | 0.27** | 0.30** | -0.01 | 0.05 | 0.06 | -0.06 | 0.02 | 0.09 |  | 0.01 | 0.02 | 0.18 | 0.17 | 0.13 | -0.15 | 0.24* | 0.19 | 0.03 | 0.09 | 0.07 | 0.14 | 0.23* |
| SL | 0.46** | 0.48** | 0.39** | 0.08 | 0.16 | 0.15 | 0.21* | 0.29** | 0.18 | 0.34** |  | 0.19 | 0.17 | 0.19 | 0.06 | 0.06 | 0.07 | -0.05 | 0.00 | 0.09 | 0.10 | 0.13 | 0.20* |
| AL | 0.06 | 0.11 | 0.11 | 0.19 | 0.14 | 0.20* | -0.18 | -0.05 | -0.13 | 0.08 | 0.09 |  | -0.03 | -0.01 | 0.06 | 0.06 | -0.10 | 0.14 | -0.04 | 0.34** | 0.34 | -0.18 | 0.07 |
| SPS | 0.40** | 0.41** | 0.40** | 0.31** | 0.32** | 0.31** | 0.18 | 0.19 | 0.13 | 0.38** | 0.55** | 0.08 |  | 0.99** | 0.33** | -0.26** | 0.47** | 0.14 | 0.01 | 0.01 | 0.04 | 0.43** | 0.47** |
| GPS | 0.42** | 0.42** | 0.39** | 0.31** | 0.44** | 0.43** | 0.18 | 0.18 | 0.15 | 0.40** | 0.55** | 0.13 | 0.96** |  | 0.34** | -0.26** | 0.47** | 0.15 | 0.01 | -0.00 | 0.00 | 0.44** | 0.47** |
| TGW | 0.52** | 0.50** | 0.60** | 0.36** | 0.48** | 0.46** | -0.05 | -0.03 | 0.12 | 0.17 | 0.38** | 0.08 | 0.38** | 0.35** |  | -0.33** | 0.59** | 0.42** | -0.03 | 0.21* | 0.22* | 0.36** | 0.54** |
| BY | -0.26** | -0.28** | -0.23* | 0.08 | -0.11 | -0.07 | -0.16 | 0.08 | -0.01 | 0.06 | -0.09 | 0.06 | -0.06 | -0.06 | -0.32** |  | -0.80** | -0.17 | 0.10 | 0.59** | 0.58** | -0.62** | -0.24* |
| HI | 0.59** | 0.62** | 0.58** | 0.24* | 0.35** | 0.39** | 0.24* | 0.18 | 0.19 | 0.43** | 0.49** | -0.04 | 0.50** | 0.48** | 0.55** | -0.48** |  | 0.36** | 0.02 | -0.18 | -0.17 | 0.83** | 0.74** |
| GPC | 0.44** | 0.44** | 0.47** | 0.36** | 0.34** | 0.47** | 0.11 | 0.18 | -0.02 | 0.15 | 0.21 | 0.14 | 0.16 | 0.15 | 0.36** | -0.36** | 0.36** |  | 0.38** | 0.32** | 0.33** | -0.15 | 0.12 |
| NHI | -0.04 | -0.09 | 0.01 | -0.01 | 0.06 | 0.09 | 0.12 | 0.06 | -0.03 | 0.06 | -0.04 | 0.13 | 0.04 | 0.03 | -0.16 | 0.28** | -0.23* | 0.29** |  | -0.17 | -0.17 | 0.32* | 0.19 |
| TNUp | 0.42** | 0.46** | 0.45** | 0.45** | 0.32** | 0.37** | 0.01 | 0.01 | 0.14 | 0.43** | 0.38** | 0.09 | 0.33** | 0.32** | 0.36** | 0.33** | 0.53** | 0.32** | -0.14 |  | 0.99** | 0.38** | 0.79** |
| NUpE | 0.43** | 0.47** | 0.46** | 0.44** | 0.31** | 0.37** | 0.03 | 0.01 | 0.14 | 0.43** | 0.39** | 0.07 | 0.32** | 0.30** | 0.36** | 0.30** | 0.53** | 0.33** | -0.13 | 0.99** |  | -0.37** | 0.32** |
| NUtE | 0.31** | 0.31** | 0.30** | 0.11 | 0.27** | 0.25* | 0.23* | 0.22* | 0.17 | 0.41** | 0.33** | -0.15 | 0.44** | 0.42** | 0.24** | -0.12 | 0.75** | -0.15 | 0.05 | 0.28** | 0.29** |  | 0.72** |
| NUE | 0.48** | 0.50** | 0.49** | 0.43** | 0.51** | 0.51** | 0.18 | 0.26** | 0.20* | 0.52** | 0.47** | -0.05 | 0.49** | 0.47** | 0.38** | 0.10 | 0.82** | 0.12 | -0.06 | 0.79** | 0.80** | 0.80** |  |

Correlation: T1- below diagonal, T2- Above diagonal

* and **: significant at the level of probability p < 0.05 and p < 0.01, respectively

CC1, CC2, CC3 & NDVI1, NDVI2, NDVI3- Chlorophyll contents and NDVI1 at booting, anthesis and grain filling stages, respectively, DH- Days to heading, DM- Days to maturity, PH- Plant height, TPM- Number of productive tiller per meter, SL- Spike length, AL- Awn length, SPS- Number of spikelet's per spike, GPS- Number of grains per spike, TGW- Thousand grain weight, BY- Biomass yield, HI- Harvest index, GPC- Grain protein content, NHI- Nitrogen harvest index, TNUp- Total nitrogen uptake, NUpE- Nitrogen uptake efficiency, NUtE- Nitrogen utilization efficiency, NUE- Nitrogen use efficiency

### 3.4 Genetic correlation estimates (rg) and predicted response from direct selection under stress condition ($CR_{T1}/R_{T1}$) for agro-morphological, physiological and NUE related traits

In the context of this study, the genetic correlations between T2 and T1 were approximately 1.0 for CC1, CC2, TSW, GY, HI, BMY and NUE. The evaluation of indirect selection efficiency under T2 concerning performance at T1 relative to the predicted response from direct selection under T1 ($CR_{T1}/R_{T1}$) showed values close to 1.0 for CC1, TSW, BMY, GPC, and NUE (Table 3). This implies that utilizing indirect selection under optimal fertilization conditions for these traits would be as effective as direct selection under reduced fertilization. However, for remaining traits, the efficiency of indirect selection would be comparatively lower than direct selection. Regarding the trait CC1, the prospects for indirect selection appeared even more favourable than direct selection ($CR_{T1}/R_{T1}$ = 1.18), mainly due to the substantially lower heritability under T1 in contrast to T2.

### 3.5 Performance of genotypes for yield and NUE related traits

Based on GY, top ten genotypes for high grain yield (S3A Fig) and poor grain yield (S3B Fig) at T1(N50) and T2 (N100) nitrogen levels were identified, respectively. The top 10 highest yielding genotypes and the least 10 poor grain yielding genotypes were identified under N50 conditions which were identified as N-insensitive (NIS) and N-sensitive (NS) genotypes,

**Table 3. Genetic correlations (rg) estimates of analysed traits at two treatments and across the years and 50 wheat genotypes and efficiency of indirect selection under T2 for performance under T1 relative to the predicted response to direct selection under T1 ($CR_{T1}/R_{T1}$) for 24 traits.**

| Traits | $r_g$ ± S.E.[a] | $CR_{T1}/R_{T1}$ |
|---|---|---|
| CC1 | 1.05 ± 0.00172 | 1.18 |
| CC2 | 0.94 ± 0.27646 | 0.89 |
| CC3 | 0.84 ± 0.50086 | 0.82 |
| NDVI1 | 0.30 ± 0.06782 | 0.32 |
| NDVI2 | 0.44 ± 0.05840 | 0.47 |
| NDVI3 | 0.63 ± 0.01239 | 0.66 |
| DH | 0.77 ± 0.97472 | 0.65 |
| DM | 0.72 ± 0.00886 | 0.73 |
| PH | 0.08 ± 0.12698 | 0.09 |
| TPM | 0.15 ± 0.01737 | 0.15 |
| SL | 0.75 ± 0.00762 | 0.83 |
| AL | 0.48 ± 0.02052 | 0.58 |
| SPS | 0.74 ± 0.00245 | 0.63 |
| GPS | 0.75 ± 0.00369 | 0.65 |
| TSW | 1.00 ± 0.00021 | 1.12 |
| GY | 0.93 ± 0.00089 | 0.82 |
| BMY | 1.02 ± 0.00057 | 1.07 |
| HI | 0.94 ± 0.01070 | 0.81 |
| GPC | 0.83 ± 0.00352 | 1.04 |
| NHI | 0.23 ± 0.02736 | 0.24 |
| TNUP | 0.70 ± 0.65115 | 0.59 |
| NUpE | 0.71 ± 0.00002 | 0.67 |
| NUtE | 0.85 ± 0.00221 | 0.92 |
| NUE | 0.93 ± 0.00001 | 0.97 |

respectively [58] Out of 10 NIS genotypes, four genotypes (UASBW13356, UASBW13358, UASBW13354 and UASBW13357) were common in the top 10 high yielding genotypes at the N100 level showing genotypic plasticity towards N50 and N100 conditions. Similarly, out of 10 NS genotypes, 7 lines (KRL237, DTW2011-56, HI8730, DL153-2, UASBW13362, WH1021 and MP1293) were also included in the 10 least yielding genotypes in T2. NUE and its associated traits also showed a wide range among genotypes under both T1 and T2 conditions (Table 4). The top 10 promising genotypes for each trait were identified under both conditions. The common genotypes in both T1 and T2 conditions for NUE (UASBW13356, UASBW13358, UASBW13354, and UASBW13357), TNUp & NUpE (UASBW13356, PBW175, Raj1972, and WH1022), NHI (UASBW13362), NUtE (UASBW13356, UASBW13358, UASBW13354, and UASBW13364) and GPC (UASBW13356, UASBW13358, PBW175, K9107, HD2967, and GW322) were identified as potential donors. Based on comparative performance of the genotypes for their NutE under T1 and T2 conditions, the genotypes were classified into four groups—viz., N efficient responsive (NER), N efficient non-responsive (NENR), N inefficient responsive (NIR) and N inefficient non-responsive (NINR) [59, 60]. Subsequently, an evaluation of their genotypic performance for NUtE was conducted under low (T1- N50) and recommended (T2-N100) N conditions (S4 Fig). The average NUtE of all 50 genotypes under T1 (30.16) and T2 condition (28.08) were considered as a cut-off for the identification of genotypic efficiency and responsiveness for N use.

Above-average genotypes under T1 were considered efficient (E) whereas below-average genotypes were considered inefficient (I). Similarly, in T2, above-average genotypes were considered responders(R), and below-average genotypes are considered non-responders (NR). In this classification, N was prefixed to indicate nitrogen. Considering together the category in T1 and T2, it was observed that efficient genotypes are higher in the utilization of absorbed N over inefficient genotypes. In this way, 20 genotypes *viz*., GW322, HD2967, HI1500, Kalyansona, KRL1-4, PBW343, UAS304, UAS323, UASBW13354, UASBW13355, UASBW13356, UASBW13357, UASBW13358, UASBW13359, UASBW13363, UASBW13364, UASBW13367, UASBW10453, UASBW12876, WH147 were categorised as efficient and responsive whereas 18 genotypes namely, HD2189, DBW14, DL153-2, DTW2011-56, HI8730, KRL237, MP1293, MP4010, NP846, PBW175, RAJ1972, RAJ4248, UASBW13360, UASBW13362, UASBW13365, UASBW10227, WH1021, and WH1022 were found as inefficient and non-responsive. Likewise, genotypes GW2013-540, HD2733, UASBW13361, UASBW12758, UASBW11948, UASBW12380, and UASBW12878, WH542, UASBW13366 were classified as efficient but non-responsive and HPW251, K9107, and UASBW12877 were categorised as in-efficient but responsive genotypes to N fertilisers.

## 3.6 Stress tolerance indices for NUE

The pooled data of grain yield in T1 and T2 for a total 50 genotypes were used to estimate different stress related indices for the identification of promising genotypes that have the least effects of stress conditions, i.e., reduced nitrogen levels (Table 4). Three different indices namely yield stability index (YSI), stress susceptibility Index (SSI), and percent reduction under stress (PR) were estimated among which higher values of YSI and lower values of SSI and PR are desirable for the identification of promising genotypes under stress conditions. Genotypes showed a mean YSI of 0.89 with a range of 0.66 to 1.05 whereas the mean SSI was 0.99 with a range of -0.44 to 2.97. PR among genotypes was 11.33 ranging from -4.97 to 33.86 among all the genotypes, exotic lines showed better performance for YSI (0.90), SSI (0.86), and PR (9.84) in comparison to indigenous Indian germplasm lines which showed mean YSI, SSI, and PR of 0.88, 1.10 and 12.5, respectively. The results indicated that two indigenous genotypes

**Table 4. Mean performance for NUE traits under T1 (soil N + 50 kg Nha$^{-1}$) and T2 (Soil N + 100 kg Nha$^{-1}$) conditions and stress indices in bread wheat genotypes.**

| S No. | Genotypes | Gr Yield (q/ha) | | NUE related traits | | | | | | | | Stress indices | | |
|---|---|---|---|---|---|---|---|---|---|---|---|---|---|---|
| | | | | NUE | | NUpE | | NUtE | | GPC | | YSI | SSI | PR (%) |
| | | T1 | T2 | T1 | T2 | T1 | T2 | T1 | T2 | T1 | T2 | | | |
| 1 | DBW 14 | 23.32 | 30.28 | 12.92 | 14.23 | 0.50 | 0.57 | 25.98 | 25.26 | 11.33 | 13.17 | 0.77 | 2.02 | 22.99 |
| 2 | DL 153–2 | 20.60 | 28.64 | 11.39 | 13.45 | 0.42 | 0.48 | 27.12 | 27.99 | 11.58 | 11.61 | 0.72 | 2.46 | 28.08 |
| 3 | DTW 2011–56 | 19.16 | 20.40 | 10.59 | 9.57 | 0.44 | 0.53 | 24.01 | 17.95 | 12.19 | 10.85 | 0.94 | 0.53 | 6.07 |
| 4 | GW 2013–540 | 28.32 | 30.33 | 15.69 | 14.24 | 0.49 | 0.54 | 32.44 | 26.91 | 11.41 | 13.06 | 0.93 | 0.58 | 6.63 |
| 5 | GW 322 | 28.78 | 31.98 | 15.96 | 15.02 | 0.50 | 0.48 | 31.84 | 31.19 | 13.04 | 13.26 | 0.90 | 0.88 | 10.00 |
| 6 | HD 2189 | 24.60 | 29.03 | 13.63 | 13.61 | 0.47 | 0.49 | 29.39 | 27.92 | 10.56 | 11.56 | 0.85 | 1.34 | 15.24 |
| 7 | HD 2733 | 29.77 | 35.54 | 16.44 | 16.66 | 0.54 | 0.63 | 30.84 | 26.60 | 12.84 | 12.78 | 0.84 | 1.42 | 16.24 |
| 8 | HD 2967 | 31.25 | 37.90 | 17.29 | 17.78 | 0.54 | 0.59 | 32.59 | 30.64 | 13.07 | 13.45 | 0.82 | 1.54 | 17.54 |
| 9 | HI 1500 | 28.94 | 30.23 | 16.01 | 14.17 | 0.53 | 0.48 | 30.73 | 32.22 | 12.32 | 12.56 | 0.96 | 0.37 | 4.26 |
| 10 | HI8730 | 19.27 | 23.34 | 10.68 | 10.97 | 0.49 | 0.52 | 22.11 | 21.24 | 11.48 | 11.28 | 0.83 | 1.53 | 17.42 |
| 11 | HPW251 | 26.88 | 30.61 | 14.88 | 14.37 | 0.51 | 0.48 | 29.11 | 30.28 | 12.74 | 13.33 | 0.88 | 1.07 | 12.17 |
| 12 | K 9107 | 31.42 | 37.29 | 17.38 | 17.49 | 0.58 | 0.57 | 30.13 | 31.11 | 13.10 | 13.63 | 0.84 | 1.38 | 15.74 |
| 13 | Kalyansona | 30.83 | 33.26 | 17.05 | 15.62 | 0.57 | 0.55 | 30.21 | 28.66 | 12.98 | 12.97 | 0.93 | 0.64 | 7.31 |
| 14 | KRL 237 | 16.87 | 25.51 | 9.36 | 11.96 | 0.40 | 0.51 | 23.41 | 23.32 | 12.16 | 12.25 | 0.66 | 2.97 | 33.86 |
| 15 | KRL1-4 | 32.15 | 34.28 | 17.81 | 16.08 | 0.58 | 0.54 | 31.06 | 29.54 | 11.85 | 12.22 | 0.94 | 0.55 | 6.21 |
| 16 | MP 1293 | 25.24 | 27.97 | 13.99 | 13.12 | 0.50 | 0.56 | 28.29 | 23.49 | 11.21 | 12.02 | 0.90 | 0.86 | 9.75 |
| 17 | MP 4010 | 26.42 | 26.72 | 14.60 | 12.54 | 0.53 | 0.58 | 27.61 | 21.78 | 12.42 | 12.17 | 0.99 | 0.10 | 1.12 |
| 18 | NP846 | 27.90 | 27.80 | 15.46 | 13.07 | 0.59 | 0.56 | 26.44 | 23.32 | 12.72 | 13.22 | 1.00 | -0.03 | -0.37 |
| 19 | PBW 175 | 32.36 | 30.83 | 17.89 | 14.49 | 0.62 | 0.62 | 28.89 | 23.31 | 12.92 | 13.63 | 1.05 | -0.44 | -4.97 |
| 20 | PBW 343 | 28.80 | 38.45 | 15.93 | 18.04 | 0.51 | 0.52 | 31.77 | 34.92 | 13.06 | 13.20 | 0.75 | 2.20 | 25.08 |
| 21 | RAJ 1972 | 29.96 | 32.89 | 16.57 | 15.41 | 0.63 | 0.62 | 26.55 | 24.96 | 12.55 | 13.70 | 0.91 | 0.78 | 8.90 |
| 22 | RAJ 4248 | 27.93 | 32.93 | 15.48 | 15.42 | 0.56 | 0.59 | 28.01 | 26.21 | 12.32 | 12.76 | 0.85 | 1.33 | 15.16 |
| 23 | UAS 304 | 33.87 | 34.43 | 18.73 | 16.15 | 0.57 | 0.53 | 32.98 | 30.64 | 11.62 | 12.20 | 0.98 | 0.14 | 1.63 |
| 24 | UAS 323 | 29.25 | 35.55 | 16.22 | 16.68 | 0.53 | 0.58 | 30.46 | 29.04 | 10.99 | 12.68 | 0.82 | 1.55 | 17.71 |
| 25 | WH 147 | 25.29 | 30.17 | 14.02 | 14.14 | 0.45 | 0.42 | 31.08 | 34.01 | 11.95 | 11.76 | 0.84 | 1.42 | 16.16 |
| 26 | WH 542 | 29.58 | 32.32 | 16.37 | 15.17 | 0.47 | 0.55 | 34.99 | 27.84 | 10.70 | 11.67 | 0.92 | 0.75 | 8.49 |
| 27 | WH 1021 | 22.83 | 26.87 | 12.61 | 12.58 | 0.47 | 0.45 | 26.99 | 27.93 | 12.21 | 12.87 | 0.85 | 1.32 | 15.04 |
| 28 | WH 1022 | 30.47 | 36.47 | 16.85 | 17.10 | 0.58 | 0.62 | 29.42 | 27.84 | 12.94 | 13.17 | 0.84 | 1.44 | 16.46 |
| 29 | UASBW13354 | 34.20 | 38.76 | 18.90 | 18.19 | 0.57 | 0.53 | 33.10 | 34.61 | 12.52 | 12.95 | 0.88 | 1.04 | 11.75 |
| 30 | UASBW13355 | 29.10 | 39.06 | 16.09 | 18.32 | 0.51 | 0.56 | 31.47 | 32.90 | 12.70 | 13.60 | 0.75 | 2.24 | 25.50 |
| 31 | UASBW13356 | 38.91 | 40.36 | 21.54 | 18.95 | 0.62 | 0.59 | 35.15 | 32.27 | 13.29 | 13.69 | 0.96 | 0.32 | 3.60 |
| 32 | UASBW13357 | 33.34 | 38.08 | 18.47 | 17.86 | 0.61 | 0.58 | 30.61 | 30.86 | 12.75 | 12.59 | 0.88 | 1.09 | 12.44 |
| 33 | UASBW13358 | 38.12 | 40.12 | 21.11 | 18.83 | 0.62 | 0.54 | 34.02 | 35.17 | 13.04 | 13.34 | 0.95 | 0.44 | 4.98 |
| 34 | UASBW13359 | 33.60 | 35.43 | 18.59 | 16.63 | 0.58 | 0.55 | 32.15 | 33.40 | 12.99 | 12.71 | 0.95 | 0.46 | 5.19 |
| 35 | UASBW13360 | 29.39 | 31.48 | 16.25 | 14.78 | 0.56 | 0.54 | 29.29 | 27.84 | 12.73 | 13.08 | 0.93 | 0.58 | 6.65 |
| 36 | UASBW13361 | 29.09 | 33.50 | 16.11 | 15.71 | 0.51 | 0.59 | 31.83 | 26.67 | 10.67 | 12.86 | 0.87 | 1.16 | 13.16 |
| 37 | UASBW13362 | 22.19 | 26.08 | 12.28 | 12.22 | 0.46 | 0.54 | 26.98 | 22.67 | 12.17 | 12.80 | 0.85 | 1.31 | 14.90 |
| 38 | UASBW13363 | 27.70 | 28.98 | 15.32 | 13.61 | 0.49 | 0.47 | 31.68 | 28.76 | 11.85 | 12.25 | 0.96 | 0.39 | 4.42 |
| 39 | UASBW13364 | 32.02 | 34.53 | 17.71 | 16.17 | 0.52 | 0.52 | 34.38 | 31.28 | 12.04 | 10.90 | 0.93 | 0.64 | 7.26 |
| 40 | UASBW13365 | 25.81 | 28.97 | 14.31 | 13.58 | 0.50 | 0.53 | 29.06 | 25.66 | 11.80 | 12.54 | 0.89 | 0.96 | 10.90 |
| 41 | UASBW13366 | 29.44 | 33.22 | 16.33 | 15.61 | 0.47 | 0.58 | 33.40 | 27.22 | 11.14 | 12.41 | 0.89 | 0.99 | 11.38 |
| 42 | UASBW13367 | 31.05 | 32.40 | 17.20 | 15.23 | 0.53 | 0.55 | 32.68 | 28.21 | 11.18 | 12.97 | 0.96 | 0.36 | 4.14 |
| 43 | UASBW12758 | 26.42 | 31.17 | 14.64 | 14.67 | 0.49 | 0.59 | 30.22 | 24.77 | 12.15 | 12.22 | 0.85 | 1.34 | 15.25 |
| 44 | UASBW10227 | 28.57 | 29.13 | 15.80 | 13.65 | 0.54 | 0.54 | 29.32 | 25.18 | 12.26 | 11.15 | 0.98 | 0.17 | 1.91 |

*(Continued)*

**Table 4.** (Continued)

| S No. | Genotypes | Gr Yield (q/ha) | | NUE related traits | | | | | | | | Stress indices | | |
|---|---|---|---|---|---|---|---|---|---|---|---|---|---|---|
| | | | | NUE | | NUpE | | NUtE | | GPC | | YSI | SSI | PR (%) |
| | | T1 | T2 | T1 | T2 | T1 | T2 | T1 | T2 | T1 | T2 | | | |
| 45 | UASBW10453 | 30.44 | 35.44 | 16.85 | 16.64 | 0.54 | 0.58 | 31.57 | 29.35 | 11.04 | 12.04 | 0.86 | 1.24 | 14.11 |
| 46 | UASBW11948 | 29.73 | 34.42 | 16.42 | 16.13 | 0.51 | 0.63 | 31.94 | 25.67 | 11.27 | 11.77 | 0.86 | 1.19 | 13.61 |
| 47 | UASBW12380 | 31.00 | 34.25 | 17.13 | 16.06 | 0.56 | 0.60 | 30.89 | 27.04 | 12.03 | 12.13 | 0.91 | 0.83 | 9.49 |
| 48 | UASBW12876 | 29.80 | 33.01 | 16.48 | 15.47 | 0.52 | 0.48 | 31.72 | 32.54 | 11.14 | 12.42 | 0.90 | 0.85 | 9.72 |
| 49 | UASBW12877 | 28.58 | 32.57 | 15.80 | 15.28 | 0.55 | 0.54 | 29.03 | 28.75 | 11.36 | 12.58 | 0.88 | 1.08 | 12.26 |
| 50 | UASBW12878 | 31.84 | 33.16 | 17.62 | 15.54 | 0.53 | 0.58 | 33.36 | 27.13 | 10.95 | 12.13 | 0.96 | 0.35 | 3.96 |
| | Mean | 28.65 | 32.32 | 15.85 | 15.16 | 0.52 | 0.54 | 30.16 | 28.08 | 12.06 | 12.56 | 0.89 | 0.99 | 11.33 |
| | CD @ 5% | 5.18 | 3.94 | 2.85 | 1.70 | 0.06 | 0.09 | 4.03 | 4.40 | 1.34 | 1.26 | | | |

NUE-Nitrogen use efficiency, NUpE-Nitrogen uptake efficiency, NUtE-Nitrogen utilization efficiency, GPC-Grain protein content, YSI- Yield stability index, SSI, Stress susceptibility index, PR- Percent reduction under stress

namely PBW175 and NP846 showed higher yield at reduced nitrogen levels which is also supported by negative SSI and PR values in PBW175 (-0.44 & -4.97) and NP846 (-0.03 & -0.37), respectively. The perusal of results of YSI, SSI, and PR indicated that PBW175, NP846, MP4010, UAS304, and UASBW10227 were promising genotypes that were least affected for grain yield under reduced nitrogen doses. In addition, the indigenous germplasm lines, lines, HI1500, DTW2011-56, KRL1-4, GW2013-540, and exotic germplasm lines UASBW13356, UASBW12878, UASBW13367, UASBW13363, UASBW13358, and UASBW13359 showed promising performance for these three stress indices.

## 4. Discussion

Nitrogen (N) is often a limiting factor for crop yield and is, therefore, considered as an essential macronutrient required for the plant growth. Globally, the adoption of N-responsive cultivars and extensive application of N fertilizer has augmented food production in the past 50 years of agriculture. This excessive application of N fertilizer is increasing the cost of cultivation and thereby reducing the net profitability at the farm level in addition to negative impacts on the environment [61–65]. The present investigation was driven by a worldwide willingness, especially from a huge number of small and marginal farmers in developing countries, to understand genotype behaviour for the identification and development of genotypes (without compromising GY) with high NUE under low N conditions. As demand for wheat grain is still rising, there is a need to boost productivity and production for the amount of N applied and therefore, identification of genotypes with better NUE is crucial. The immediate aim should, therefore, be to exploit the available variability for NUE in wheat cultivars through an appropriate breeding procedure.

The present study aimed to understand the variability and correlations for NUE traits with various yield and physiological parameters to identify the promising genotypes based on various parameters of nitrogen sensitivity, response to N fertilizers, and susceptibility indices at low N availability. Précised phenotyping under low N input is challenging and influenced by the genotype (G), environment (E), and G × E interactions [66, 67] which makes it difficult to identify nitrogen insensitive genotypes under low N conditions at field level. Presently, very few wheat breeding programs are targeting the development of low N tolerance traits which are a must for sustainable agriculture with minimal negative impacts on the environment.

Analysis of variance indicated a highly significant variation among genotypes at both treatments for all the traits studied and this existence of significant differences for different

attributes is very helpful to improve the trait of interest. Diverse responses have been observed among genotypes for all traits across N levels, despite similar growth conditions and an equal amount of N fertilizer in a given N level. The genotypic variations observed purely reveal trait specific genotype plasticity. These results concur with other reported field experiments on wheat [68, 69].

Wheat is more responsive to a higher dose of nitrogen whereas, sensitive to low nitrogen availability like most cereal crops. Less soil N availability leads to a reduced plant vegetative growth especially tillering which results in reduced grain yield [70]. In this study, reduction in tillering and ultimately yield reduction to the tune of approx. 11.3% was observed at reduced N levels as compared to the recommended dose. Similarly, the low N availability is expressed phenotypically in the form of low chlorophyll content as indicated by lower SPAD and NDVI values which might be due to less N availability in the leaves at lower N levels. Similar results of increased chlorophyll content by increasing N application were reported by [71–73]. Almost all the traits studied except NDVI-1, HI, NUE, NUtE and NHI, showed reduced performance at lower N doses which reflects the genotypic sensitivity to low N availability [74]. When N supply was low, the remobilization of N from the tillers and leaves was efficient and converted into grain N indicating a better NUtE. This suggested that few genotypes have the potential to use nitrogen more efficiently and exhibiting high NUE. The NHI indicates the level of efficiency of plants to use acquired total N for grain formation for which significant genotypic variation was observed at both the N levels [75]. Varied response of genotypes to NUpE, NUtE, and NHI was observed by [28, 33, 76–79] which are in tune with the performance of NUE related traits at varying N levels in the present study. From per cent reduction calculations we came to know that a significant reductions were observed in almost traits at T1 in comparison to T2 except, NHI (2.8%), NUtE (7.4%), and NUE (4.6%) traits which exhibited notably higher mean values at T1 when contrasted with T2. A study was also conducted on similar investigation, revealing a 10% reduction in yield and a 14% decrease in protein content at LN as opposed to HN conditions [80]. Additionally, grain N yield and aboveground N per unit area were both reduced by approximately 20% under LN conditions.

To assess the nature and magnitude of diversity among the genotypes, the phenotypic and genotypic coefficients of variation were estimated. In general, higher PCV values were observed as compared to GCV values at both the nitrogen levels. High PCV and GCV were observed for harvest index whereas other major yield traits and NUE related traits showed moderate values. Similarly, previous studies have established significant genetic variation for NUE-related traits in wheat [81, 82].

In general, moderate to high heritability was recorded under both nitrogen levels. The heritability values in a broad sense ranged from 34.05% to 86.84% in T1 and 37.81% to 93.84% in T2. High heritability (>60%) were observed for CC-3, DH, DM, SPS, GY, BMY, HI, NUE, and NUtE under both conditions. A similar trend was also observed for yield and NUE related traits by [83, 84]. High heritability is useful in laying importance in choice for such traits during selection.

In the present study, nitrogen use efficiency was measured as grain yield divided by available N (soil N + fertilizer N) due to which the correlation of NUE and grain yield at both levels of nitrogen treatments showed similar results. The positive and highly significant correlated response of NUE with DM, HI, TGW, GPS, SPS, CC-1, CC-2, CC-3, NDVI-1, NDVI-2, NDVI-3 TNUp, NUpE, and NUtE under both the condition suggested improvement of these traits for better NUE in wheat genotypes [85, 86]. Interestingly, highly significant and positive associations between physiological traits and NUE related traits were also observed. The results of BLUP analysis also indicated reliability of NUE related traits with positively moderate values for improvement for grain yield. Similar and contrast regression relationships from BLUP

analysis were also graphically depicted involving grain yield vs GPC, grain yield vs NUpE, and grain yield vs NUtE by [80], Similar associations of NUE related traits by [46, 70, 79, 87, 88] indicated that these characters should be considered as NUE components in crop breeding that can be harnessed for higher grain yields under both low and high N fertility conditions. The current study highlighted that genetic correlations between T2 and T1 were notably high for CC1, CC2, TSW, GY, HI, BMY, and NUE, indicating the absence of significant genotype by nitrogen (G x N) interactions for these traits. Heritability estimates for the assessed traits either showed similarity across both N treatment levels or were higher at T2 compared to T1. Reflecting this, the indirect selection efficiency (ISE) at T2 for performance at T1, relative to direct selection at T1, was equal to or greater than 1 for CC1, TSW, BMY, GPC, and NUE. This aligns with previous findings from studies by [80, 89], underscoring its significant implications for the selection of wheat cultivars with enhanced nitrogen efficiency.

Across the year on average, higher GY was recorded at higher N doze and large genetic variability among genotypes for GY was observed. In general, increased GY was correlated with the enhanced N application rate, which might be due to sufficient nitrogen availability. Reduced GY by 10% under low N conditions compared to normal conditions was also reported by [65, 89] in wheat. Remarkably, top 10 highest yielding genotypes were identified under N50 conditions as N-insensitive (NIS) among which four genotypes (UASBW13356, UASBW13358, UASBW13354, and UASBW13357) were common in the top 10 high yielding genotypes at the N100 level showing their genotypic plasticity towards varying N levels. These genotypes may be potential genetic resources to breed for tolerance to low N conditions. On the other hand, the top 10 high yielding genotypes in N100 can have the best acceptances for cultivation where soils are fertile and following the ideal N levels for cultivation and could be further used to improve GY, along with higher NUE. Several studies have suggested the utilization of N efficient lines with enhanced GY in the farmer fields which may help to reduce fertilizer input as well as increase the profitability of farm operations [14, 75, 90]. Similarly, selected NIS lines with high NUE will certainly play a role in reducing environmental pollution and could increase economic profit for farmers.

The present study indicated that NUE and its associated traits also showed a wide range among genotypes for all the six NUE related traits and their diagrammatic representation of frequency distributions under T1 and T2 conditions were depicted in S5 Fig. The higher frequency of genotypes in medium to high class based on NUE parameters suggests the potentiality of these tested genotypes. The best performing 10 entries were identified for each trait under T1 and T2 conditions among which common genotypes under both the conditions were identified as promising trait specific genetic resources. Further, UASBW13356, UASBW13358, UASBW13354, PBW 175, Raj 1972, and WH 1022 were found promising for multiple NUE traits and may be utilized extensively for NUE improvement programme in wheat. Determination of the genetic variations for NUE related traits is essential for the selection of efficient genotypes that can be used further in breeding programs to develop low N tolerant material. The concept of genotype grouping is used widely in nutrient use efficiency [91]. Among NUE related traits, nitrogen utilization efficiency has the most significance which is defined as the genotype ability to assimilate and remobilize N ultimately to produce the GY [16, 59]. Large genotypic variations in NUtE have been reported under field/pot screening in various wheat genotypes and other crops [92–94]. Based on NUtE efficiency data in T1 (N50) versus T2 (N100), above and below average genotypes under T1 were considered efficient and inefficient, respectively whereas, above and below average genotypes in T2 were considered as responders, and non-responders, respectively.

Based on NUtE in T1(N50), 29 efficient (E) and 21 inefficient (I) genotypes were identified whereas 23 responsive (R) and 27 non-responsive (NR) genotypes were identified based on

NUtE in T2 (N100), Considering together, all the 50 genotypes were classified into four groups viz., NER (20), NENR (9), NIR (03) and NINR (18) as per [59, 60]. The NENR genotypes showed a progressive performance under low N which may enable breeders to develop efficient genotypes under low input environments. The remaining 20 NER genotypes were most desirable that exhibited a progressive response to increased N availability. These NER genotypes could be the prospective targets for selection for the genetic improvement of wheat for better N utilization. Interestingly, among the 10 NIS genotypes, 8 genotypes namely, UASBW13356, UASBW13358, UASBW13354, UAS304, UASBW13359, UASBW13357, KRL1-4, and UASBW13364 were N efficient and responsive which showed progressive performance in terms of efficient and responsive use of nitrogen. High GY per unit of N consumption could be obtained from the genotypes having more NUtE [19], and thus, breeding for efficient genotypes under low N could be progressed with high NUtE [75, 95, 96]. Similarly, inefficient but responsive genotypes (HPW251, K9107, and UASBW12877) can be used in breeding programs for further improvement. The rest of the inefficient, nonresponsive genotypes are less desirable from the NUE point of view. Overall, it was found that the efficient genotypes are higher in the utilization of absorbed nutrients than inefficient genotypes.

The differential behaviour of genotypes under stress was studied by estimation of three different stress related indices for the identification of promising genotypes that have the least effects of stress conditions, *i.e.*, reduced nitrogen levels. Among these indices, higher values of YSI and lower values of SSI and PR were desirable for the identification of promising genotypes under stress conditions. A wide range was observed among genotypes for these indices and exotic genotypes showed more desirable indices. Based on these indices, the indigenous germplasm lines, HI1500, DTW2011-56, KRL1-4, GW2013-540, and exotic germplasm lines UASBW13356, UASBW12878, UASBW13367, UASBW13363, UASBW13358, and UASBW13359 were found promising. These indices also indicated PBW175, NP846, MP4010, UAS304, and UASBW10227 as the least affected genotypes for grain yield under reduced nitrogen doses.

## 5. Conclusions

Improving NUE is pivotal for a sustainable crop growth and yield, especially under low nitrogen soils on which depend majority of poor farmers in most developing countries, especially in South Asia. Likewise, improving crop productivity using N fertilization is important for achieving climate resilience. Nevertheless, the genetic improvement of NUE depends on the nature and extent of variation among the germplasm. This study aimed to derive morphologic and agronomical traits associated with NUE in a set of 50 diverse lines under different N levels. Although, the nitrogen limitation has resulted in the reduction of GY, extensive genetic variations for grain yield, NUE, and their associated traits were noted among genotypes at both levels of nitrogen. The association between yield and NUE traits indicated avenues for wheat improvement concerning enhanced NUE and thereby, yield. Different selection parameters for the selection of nitrogen-insensitive genotypes, nitrogen efficient and responsive genotypes, and stress tolerance due to low N availability have identified UASBW13356, UASBW13358, UASBW13354, UASBW13357, and KRL1-4 as promising genotypes that showed their inherent genotypic plasticity toward N application. These findings suggest that genotypes with more yield and high to moderate NUtE can be used as parents for the breeding of N efficient genotypes for marginal agro-ecologies. Furthermore, these nitrogen-use efficient genotypes can be further tested on large scale to know their stability for their release as a variety so that farmers can get the advantage of low input cost. These genotypes have the potential to decrease environmental pollution and high economic costs associated with excess N fertilizer. Additionally, these are the best suggested genotypes for organic farming due to their

inherent potentiality of low input response which may also help the vast number of small and marginal farmers for optimizing the use of fertilizer inputs for economic and environmentally sustainable food production.

## Supporting information

**S1 Fig. The relationship between grain yield and NUE under two nitrogen levels.** The grain yield and NUE under the two nitrogen levels N50 and N100 is showing similar trend as NUE is estimated by dividing grain yield available N soil.
(TIF)

**S2 Fig. BLUP analysis for correlation of grain yield with NUE traits.** GY showed (D) perfectly positive correlation with NUE, positive but weak correlation with (A) GPC and (B) NUpE under both N levels. GY showed (C) moderately positive correlation under T1 and weaker but positive correlation under T2 with NUtE.
(TIF)

**S3 Fig. Top high yielding and poor yielding genotypes under T1 and T2 conditions.** Top ten genotypes for high grain yield (A) and poor grain yield (B) at T1(N50) and T2 (N100) nitrogen levels were identified.
(TIF)

**S4 Fig. Relationship between genotypic performance for NUtE under low (T1-N50) and recommended (T2-N100) nitrogen conditions.** The genotypes were classified into four groups—viz., N efficient responsive (NER), N efficient non-responsive (NENR), N inefficient responsive (NIR) and N inefficient non-responsive (NINR) based on comparative performance of the genotypes for their NutE under T1 and T2 conditions.
(TIF)

**S5 Fig. Frequency distribution for various NUE related traits in wheat under T1 and T2 conditions.** Frequency distribution pattern for all the six NUE related traits namely TNUp, NUtE, NUpE, NUE, NHI and GPC was studied under T1 and T2 conditions.
(TIF)

**S1 Table. Mean performance of bread wheat genotypes for agro-morphological, physiological and NUE traits under $T_1$ (soil N + 50 kg Nha$^{-1}$) and T2 (Soil N + 100 kg Nha$^{-1}$) conditions. DH:** Days to Heading, DM: Days to maturity, PH: Plant height (cm), NPT: Number of productive tiller per meter, SL: Spike length (cm), AL: Awn length (cm), SPS: spikelet's per spike, GPS: grains per spike, TSW: Thousand seed weight (g), GY: Grain yield (q ha$^{-1}$), BMY: Biomass yield (q ha$^{-1}$), HI: Harvest index (%), GPC: Grain protein content (%), CC-1: Chlorophyll content at booting stage, NDVI -1: NDVI at booting stage, CC-2: Chlorophyll content at anthesis stage, NDVI- 2: NDVI at anthesis stage, CC-3: Chlorophyll content at grain filling stage, NDVI -3: NDVI at grain filling stage, NHI: Nitrogen harvest index, TNUp: Total nitrogen uptake (kg N ha$^{-1}$), NUpE: Nitrogen uptake efficiency (kg N kg$^{-1}$N), NUtE: Nitrogen utilization efficiency (kg grain kg$^{-1}$N) and NUE: Nitrogen use efficiency (kg grain kg$^{-1}$N).
(DOCX)

## Acknowledgments

We are thankful to the technical staff at Dr Sanjaya Rajaram Wheat Research Laboratory for data collection at the experimental site and laboratory technicians for Nitrogen (N) analysis in different samples.

## Author Contributions

**Conceptualization:** Mahalaxmi K. Patil, S. A. Desai, V. Rudra Naik.

**Data curation:** Suma S. Biradar, Mahalaxmi K. Patil.

**Formal analysis:** Suma S. Biradar, Mahalaxmi K. Patil, Sanjay K. Singh.

**Investigation:** Mahalaxmi K. Patil, Kumar Lamani.

**Methodology:** Suma S. Biradar, Mahalaxmi K. Patil, S. A. Desai, Kumar Lamani.

**Resources:** Arun K. Joshi.

**Supervision:** S. A. Desai.

**Visualization:** Suma S. Biradar, V. Rudra Naik, Arun K. Joshi.

**Writing – original draft:** Suma S. Biradar, Mahalaxmi K. Patil.

**Writing – review & editing:** S. A. Desai, Sanjay K. Singh, V. Rudra Naik.

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
