## [Decision Letter · Decision Letter 0]

13 Jun 2023

PONE-D-23-04727Nitrogen Use Efficiency in Bread Wheat: Genetic Variation and Prospects for Improvement PLOS ONE

Dear Dr. Desai,

Thank you for submitting your manuscript to PLOS ONE. After careful consideration, we feel that it has merit but does not fully meet PLOS ONE’s publication criteria as it currently stands. Therefore, we invite you to submit a revised version of the manuscript that addresses the points raised during the review process.

Revise the manuscript as per the reviewers' comments

We look forward to receiving your revised manuscript.

Kind regards,

Vijay Gahlaut, Ph.D.

Academic Editor

PLOS ONE

Reviewers' comments:

Reviewer's Responses to Questions

**Comments to the Author**

1. Is the manuscript technically sound, and do the data support the conclusions?

Reviewer #1: Yes

Reviewer #2: Yes

2. Has the statistical analysis been performed appropriately and rigorously? 

Reviewer #1: Yes

Reviewer #2: Yes

3. Have the authors made all data underlying the findings in their manuscript fully available?

Reviewer #1: Yes

Reviewer #2: No

4. Is the manuscript presented in an intelligible fashion and written in standard English?

Reviewer #1: Yes

Reviewer #2: No

5. Review Comments to the Author

Reviewer #1: This is an interesting paper and topic. Overall, it's well done but i encourage the authors to eliminate repetitive text and to focus on being concise with the results section. Not everything that is significant in every table needs to be highlighted. It would be more useful if the authors focused less on this and more on why differences may occur - an explanation.

Reviewer #2: REVIEWERS COMMENTS

NEEDS MAJOR REVISION PRIOR ACCEPTANCE

1. There are many grammatical errors of prepositions, verbs etc. Please check the pdf file attached for detailed corrections

2. Have you added any supplementary file along with the main manuscript? As under methodology section, like in case of Table 3, you have mentioned few traits, the tables containing values for the rest of traits are missing.

3. Statistical part is fine but please also refer to this article for statistical part analysis Ivić M, Grljušić S, Plavšin I, Dvojković K, Lovrić A, Rajković B, Maričević M, Černe M, Popović B, Lončarić Z, Bentley AR, Swarbreck SM, Šarčević H and Novoselović D (2021) Variation for Nitrogen Use Efficiency Traits in Wheat Under Contrasting Nitrogen Treatments in South-Eastern Europe. Front. Plant Sci. 12:682333. doi: 10.3389/fpls.2021.682333

4. Add latest references of 2022-2023 as well related to nitrogen use efficiency in wheat.

5. Check for the spelling of this reference in the text. Choukan, R., Taherkhani, T., Ghanadha, M.R., Khodarahmi, M., 2006. Evaluation of drought tolerance in grain maize inbred lines using drought tolerance indices. Iranian Journal of Crop Sci. 8(1), 79-89.

6. PLOS authors have the option to publish the peer review history of their article (what does this mean?). If published, this will include your full peer review and any attached files.

Reviewer #1: No

Reviewer #2: **Yes: **MOHINI PRABHA SINGH

---

## [Author Response · Author response to Decision Letter 0]

16 Aug 2023

Corrections are made in the manuscript and also highlighted in the track-changes version of the Manuscript.

---

## [Decision Letter · Decision Letter 1]

9 Nov 2023

Nitrogen Use Efficiency in Bread Wheat: Genetic Variation and Prospects for Improvement 

PONE-D-23-04727R1

Dear Dr. Desai,

We’re pleased to inform you that your manuscript has been judged scientifically suitable for publication and will be formally accepted for publication once it meets all outstanding technical requirements.

Kind regards,

Vijay Gahlaut, Ph.D.

Academic Editor

PLOS ONE

Additional Editor Comments (optional):

Reviewers' comments:

Reviewer's Responses to Questions

**Comments to the Author**

1. If the authors have adequately addressed your comments raised in a previous round of review and you feel that this manuscript is now acceptable for publication, you may indicate that here to bypass the “Comments to the Author” section, enter your conflict of interest statement in the “Confidential to Editor” section, and submit your "Accept" recommendation.

Reviewer #2: All comments have been addressed

2. Is the manuscript technically sound, and do the data support the conclusions?

Reviewer #2: Yes

3. Has the statistical analysis been performed appropriately and rigorously? 

Reviewer #2: Yes

4. Have the authors made all data underlying the findings in their manuscript fully available?

Reviewer #2: Yes

5. Is the manuscript presented in an intelligible fashion and written in standard English?

Reviewer #2: Yes

6. Review Comments to the Author

Reviewer #2: As per the suggestions asked from authors, they have incorporated all the necessary suggestions. I find the manuscript fit for acceptance.

1. Please send the corrected supplementary file as well

2. Ivić M, Grljušić S, Plavšin I, Dvojković K, Lovrić A, Rajković B, Maričević M, Černe M, Popović B, Lončarić Z, Bentley AR, Swarbreck SM, Šarčević H and Novoselović D (2021) reference is added in analysis

3. Recent references of 2022-23 in the Manuscript were found to be added.

4. The spelling of this reference , Choukan, R., Taherkhani, T., Ghanadha, M.R., Khodarahmi, M., 2006. Evaluation of drought tolerance in grain maize inbred lines using drought tolerance indices. Iranian Journal of Crop Sci. 8(1), 79-89 is corrected.

7. PLOS authors have the option to publish the peer review history of their article (what does this mean?). If published, this will include your full peer review and any attached files.

Reviewer #2: **Yes: **MOHINI PRABHA SINGH

---

## [Editor Report · Acceptance letter]

28 Jan 2024

PONE-D-23-04727R1 

PLOS ONE

Dear Dr. Desai, 

I'm pleased to inform you that your manuscript has been deemed suitable for publication in PLOS ONE. Congratulations! Your manuscript is now being handed over to our production team.

Kind regards, 

on behalf of

Dr. Vijay Gahlaut 

Academic Editor

PLOS ONE